# Deciphering the combinatorial landscape of immunity

**Antonio Cappuccio[1,2], Shane T Jensen[3], Boris M Hartmann[2], Stuart C Sealfon[2], Vassili Soumelis[1,4]\*, Elena Zaslavsky[2]\***

[1]Institut Curie, Integrative Biology of Human Dendritic Cells and T Cells Laboratory, PSL Research University, Inserm, U932, Paris, France; [2]Department of Neurology, Icahn School of Medicine at Mount Sinai, New York, United States; [3]Department of Statistics, Wharton School, University of Pennsylvania, Philadelphia, United States; [4]Laboratoire d'immunologie, biologie et histocompatibilité, AP-HP, Hôpital Saint-Louis, Paris, France

**Abstract** From cellular activation to drug combinations, immunological responses are shaped by the action of multiple stimuli. Synergistic and antagonistic interactions between stimuli play major roles in shaping immune processes. To understand combinatorial regulation, we present the immune Synergistic/Antagonistic Interaction Learner (iSAIL). iSAIL includes a machine learning classifier to map and interpret interactions, a curated compendium of immunological combination treatment datasets, and their global integration into a landscape of ~30,000 interactions. The landscape is mined to reveal combinatorial control of interleukins, checkpoints, and other immune modulators. The resource helps elucidate the modulation of a stimulus by interactions with other cofactors, showing that TNF has strikingly different effects depending on co-stimulators. We discover new functional synergies between TNF and IFNβ controlling dendritic cell-T cell crosstalk. Analysis of laboratory or public combination treatment studies with this user-friendly web-based resource will help resolve the complex role of interaction effects on immune processes.

**\*For correspondence:**
vassili.soumelis@aphp.fr (VS);
elena.zaslavsky@mssm.edu (EZ)

**Competing interests:** The authors declare that no competing interests exist.

## Introduction

Immunological processes are regulated by a broad diversity of stimuli, such as growth factors, cytokines, integrins, and pathogen components. Studies of immune regulation have largely focused on individual stimuli, elucidating transcriptional and functional programs induced by major immune modulators such as TNF (*Locksley et al., 2001*; *Kalliolias and Ivashkiv, 2016*; *Bouwmeester et al., 2004*), interferons (*Pestka, 2007*; *Schneider et al., 2014*; *González-Navajas et al., 2012*), and Toll-like Receptor (TLR) ligands (*Kawasaki and Kawai, 2014*; *Vidya et al., 2018*). However, immunological responses are constantly shaped by the combined action of multiple stimuli. As a fundamental example, T helper (Th) cell activation requires a combination of antigen presentation and costimulatory molecules (*Smith-Garvin et al., 2009*). Due to combinatorial effects, the study of individual stimuli alone may provide a misleading oversimplification of their role in complex microenvironments. Developing a systematic understanding of how combinations of stimuli control the immune system is an unmet challenge, with broad biological and therapeutic implications.

What makes studying immunology from a combinatorial perspective essential is that nonlinear interactions often make the effects of combined stimuli dramatically different from merely additive effects seen with each stimulus alone. Immunological studies have discovered that interaction effects play major roles in shaping immune system processes (*Noubade et al., 2014*; *Min et al., 2012*; *Tuvim et al., 2012*; *Walasek et al., 2012*; *Hartmann et al., 2014*; *Teles et al., 2013*). However, the complexity of these effects and the lack of resolution of current combinatorial experiment analysis methods are a barrier to understanding how interactions regulate immunological functions. The

standard approach to this problem is to identify genes showing positive (synergistic) or negative (antagonistic) deviations in the combined treatment condition from the sum of individual effects. This restricted binary classification has critical limitations. First, it does not distinguish between biologically relevant and artifactual interactions that depend on the specific assay (*Cappuccio et al., 2015*). Second, it does not allow for meaningful aggregation of interactions to delineate coherent functional programs. Finally, this standard approach does not support integration and shared interpretation of interactions from multiple combination treatments. These limitations obscure general properties and functions of interactions between immunological stimuli.

To overcome this barrier, we develop a framework based on combinatorial analysis and machine learning. Using combinatorial analysis, we derive a comprehensive taxonomy of all interaction patterns that can occur in a combination treatment experiment. Next, we train a machine learning classifier able to robustly map noisy -*omics* data across the pre-defined taxonomy. As we demonstrate, this framework now resolves the most informative synergistic and antagonistic effects. Annotation of genes assigned to each discrete regulatory pattern identifies the regulated immunological functions. In addition, assignment of responses from multiple combinatorial datasets to a standardized taxonomy enables global integration to systematically elucidate the interactions that shape immune functions.

The resulting immune Synergistic and Antagonistic Interaction Learner (iSAIL) resource includes the machine learning classifier, a curated compendium of immunological combination treatment datasets and a global dataset integration to produce a landscape of ~30,000 interactions from diverse immune cell types and combinations of stimuli. We show how the landscape can be mined to reveal new insight into the prevalence of different types of interactions, and into their immunological functions. Further, we show that the resource facilitates the discovery of new combinatorial regulatory processes. iSAIL analysis of data on TNF and IFNβ co-treatment led us to discover new synergistic mechanisms implicated in dendritic cell-T cell crosstalk.

iSAIL is available as a user-friendly web-accessible resource to help elucidate the combinatorial complexity of immunity.

## Results

### Background and motivation

The prototypical combination treatment experiment comprises -*omics* data obtained from the following conditions: a vehicle control (denoted by 0), two individual stimuli (X, Y), and their combination (X+Y), *Figure 1A*, left subpanel. The goal of this experiment is to discover cellular processes regulated by positive (synergistic) and negative (antagonistic) interactions. This requires a fine-grained analysis able to distinguish between biologically relevant and artifactual interactions, and able to aggregate similar interactions to delineate coherent functional programs. The standard approach, which classifies interactions as positive or negative, does not do this adequately, leading to a loss of important biological information. For example, biologically meaningful positive interactions such as emergent responses (e.g. an increase or decrease with the combination in the absence of any effect in the individual treatment conditions), would not be distinguished from nominal interactions with low biological significance related to saturation effects. Furthermore, the aggregation of all the positive (or negative) interactions obscures the inference of coherent biological processes as typically obtained through pathway enrichment analysis, because this crude aggregation confounds very diverse interaction effects.

### Machine learning framework to map interaction effects from -omics data from combination treatment studies

To improve the extraction of biological insight from a combination experiment, we develop a machine learning framework within the immune Synergistic and Antagonistic Interaction Learner (iSAIL) resource. The framework increases the resolution of a standard approach by using an abstract, pre-defined taxonomy of 123 distinguishable response profiles that can theoretically occur in a combination treatment experiment (*Figure 1—figure supplement 1*). These profiles represent qualitatively different scenarios for the expression of a gene in a combination treatment experiment. To analyze data on combination treatment, iSAIL applies a machine learning classifier trained to

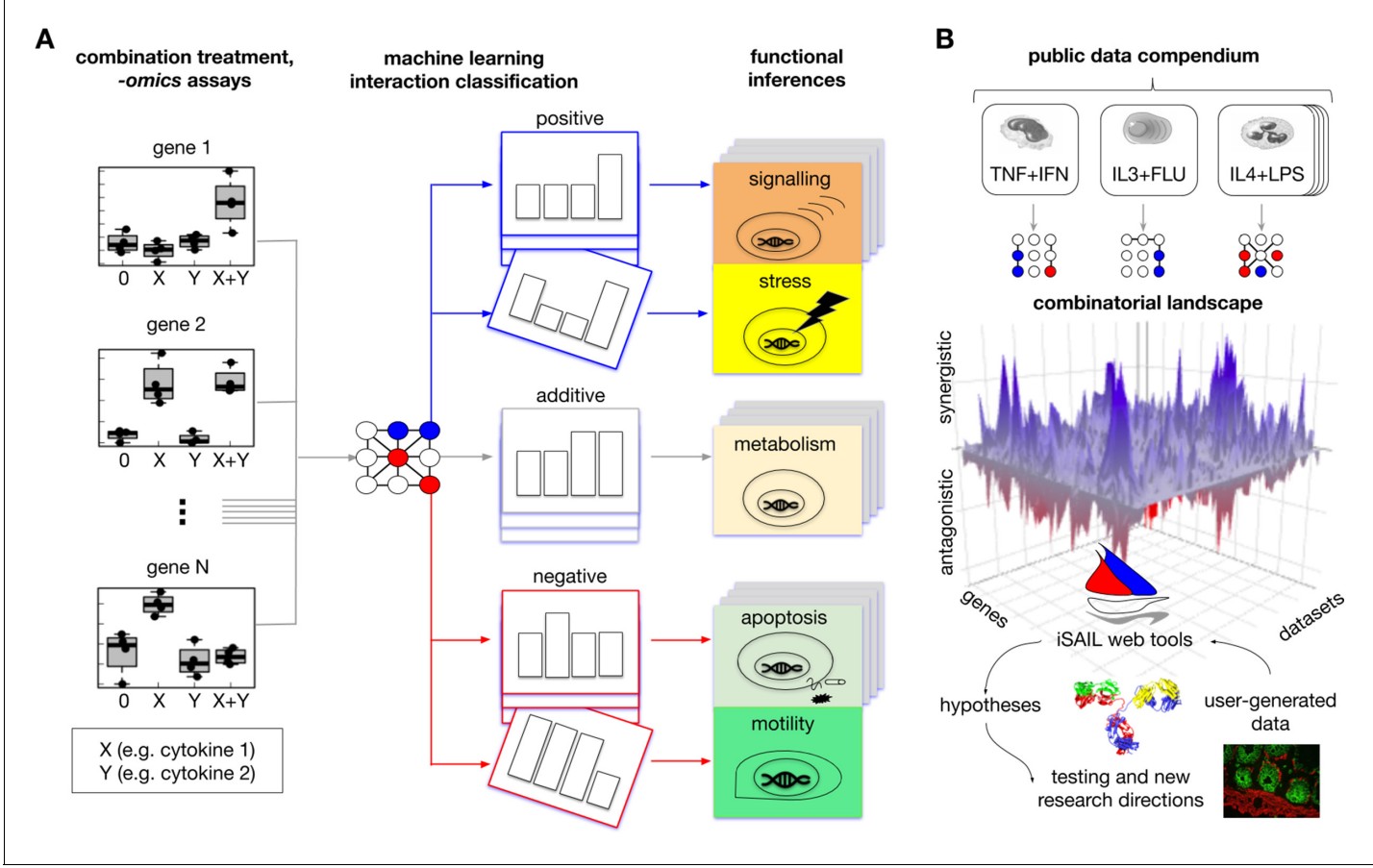

**Figure 1.** Comprehensive approach to map interaction effects within and among -*omics* datasets from combination treatment experiments. (**A**) We developed iSAIL (immune Synergistic/Antagonistic Interaction Learner), a machine learning framework to decipher the effect of combination treatments. iSAIL accommodates -*omics* datasets from the four prototypical conditions of combination treatments: 0 (control), stimulus X, stimulus Y, and the combination X+Y (left subpanel). The dataset is analyzed by a classifier previously trained to map each gene into a complete taxonomy of theoretically possible interaction profiles (middle subpanel, see also *Figure 1—figure supplement 1*). The slanting of a single profile rendition in the middle subpanel is to indicate that the slanted pattern is one of an entire deck of synergistic or antagonistic patterns classified by iSAIL. The taxonomy helps infer the functional role of different types of positive and negative interactions (right subpanel). (**B**) By applying iSAIL to combination treatments from diverse immune cell types (right subpanel), we built a combinatorial landscape of immunity comprising ~30,000 interactions. Global analysis of the landscape and of user-generated data drive new hypotheses on the role of interactions in immune cells.

The online version of this article includes the following figure supplement(s) for figure 1:

**Figure supplement 1.** A comprehensive taxonomy of response profiles to combination treatments.
**Figure supplement 2.** Machine learning classification of treatment interactions.
**Figure supplement 3.** Distribution of precision and recall of LDA and RF across the taxonomy of response profiles.

robustly classify the experimental gene response across the taxonomy (*Figure 1A*, middle subpanel). Interaction classification makes it possible to resolve the most meaningful interactions (e.g. emergent responses), while automatically excluding from downstream analysis all the nominal interactions (e.g. saturation effects) that carry minimal biological information. Importantly, genes classified in the same interaction profile are likely to share a coherent functional program. These coherent gene groups are then interrogated for mechanistic insight via pathway enrichment analysis (*Figure 1A*, right subpanel). By breaking down the combinatorial effects into these functional units, we greatly improve the functional interpretation of biological processes regulated by interactions.

A key problem we address is to robustly classify gene response from noisy -*omics* data across. To solve this problem, the classifier generates a probabilistic assignment of the response for each gene to the taxonomy-defined profiles. This solution is more robust to noise than the deterministic matching previously used for related problems (*Cappuccio et al., 2015*). We systematically validate our

approach by evaluating the classification performance on simulated data. Furthermore, in order to select the optimal framework, we investigate multiple machine learning models and compare their probabilistic output using the synthetic data under different noise regimes. These analyses identify Random Forest as a robust method with high predictive accuracy across all noise regimes (*Figure 1—figure supplements 2* and *3*).

## A resource to analyze user-generated or public immunological combination treatment datasets

The taxonomy of interaction profiles used by the learning framework provides a standardized abstract space, independent of any specific dataset, where interactions from multiple datasets can be mapped and integrated for global analysis. Based on immunological relevance and data quality (see Materials and methods), we select a compendium of 25 human and 7 murine combination treatment studies for inclusion in the resource (*Table 1*). These datasets include studies of innate immune cells including several dendritic cell populations, macrophages, and monocytes, as well as epithelial cells and adaptive immune cells. Stimuli include diverse combinations of cytokines, Toll-Like Receptor (TLR) ligands, and drugs. We apply our classification framework to each dataset from the compendium and agglomerate the results in an overall interaction landscape, including it as a component of the resource (*Figure 1B*).

The implemented interaction classification methodology, along with the interaction landscape, is made available in a user-friendly platform. The platform allows researchers to upload and analyze interactions from newly generated data, or to mine the interaction landscape, derived from our compendium, for global insight. To facilitate the functional interpretation of interactions, the platform is integrated with external resources including ImmPort (*Bhattacharya et al., 2014*), Gene Cards (*Stelzer et al., 2016*), and enrichR (*Kuleshov et al., 2016*), that provide extensive single-gene and pathway level annotations. The iSAIL resource has the capacity to generate insight into combinatorial immunity and help guide hypothesis generation and further experimentation.

## Exploring and visualizing the interaction landscape

To build the interaction landscape, we systematically applied our interaction classification framework (*Figure 2A*) to the data compendium, mapping a total of 29,479 interactions. The number of interactions vary widely with the different cell types and combinations of stimuli (*Figure 2—figure supplement 1*). The landscape, extensively annotated using external databases, is an information-rich object that can be mined to gain new insight on general properties of interactions and on the immunological functions they regulate.

For example, mining the landscape enables a systematic quantification of the prevalence of interactions that may occur with combination treatments, which is currently unknown. We find that the most frequent interactions likely represent range limitations of the assay or biological responses, which we refer to as floor and ceiling effects (*Figure 2B*, top subpanel). These effects provide limited biological information in the absence of dose-response curves (*Chou, 2006*), unavailable in our context. Our approach segregates floor and ceiling effects from less frequent but more biologically relevant interactions, including suppression (9%), inhibition (8%), restoration (4%), emergence (4%), and potentiation (3%) (*Figure 2B*, middle subpanel). Our analysis also reveals that several theoretically possible combinatorial effects were nearly absent (<0.04%). These rare profiles include reversals, where two signals with the same individual effect (e.g. upregulation of a gene separately by X and Y) are reversed by the combination (e.g. downregulation of the same gene by X+Y) (*Figure 2B*, bottom subpanel).

To explore the immunological functions regulated by interactions, we visualize the landscape using an interactive 3D representation. The three axes represent datasets, genes, and scores quantifying the intensity and robustness of the identified interactions (*Figure 2C*). Each axis is then annotated with additional information. The dataset axis is annotated with metadata on the experiments including species, cell type, stimuli, and time point. The gene axis is annotated with immunological gene families such as chemokines, interleukins, checkpoints and other terms from the ImmPort database (*Bhattacharya et al., 2014*). The interaction axis is annotated by the taxonomy profiles assigned to each interaction by the machine learning classifier. Slicing the landscape along specific dimensions provides different types of insight into immunological interactions.

**Table 1.** Datasets used to construct the combinatorial landscape of immunity.

| Accession | Species | Cell type | Signal X | Signal Y | Time point |
|---|---|---|---|---|---|
| GSE5054 | Human | Thyroid cells | IFNγ | IL1β | 1d |
| GSE36331 | Human | ARPE-19 cells | IFNγ | TNF | 2d |
| GSE43409 | Human | Innate lymphoid cells | cocktail (IL-1/IL-7/IL-23) | aNKp44 | 3.5 hr |
| GSE53712 | Human | Monocytic THP-1 | LPS | SB203580 | 4h |
| GSE53712 | Human | Monocytic THP-1 | LPS | SB203580 | 1d |
| GSE59179 | Human | Hut78 cells | Enzastaurin | AR-A014418 | 3d |
| GSE63038 | Human | NK cells | FcR activation | IL-12 | 12 hr |
| GSE79077 | Human | MDMs | Dexamethasone | IFNγ | 20 hr |
| GSE57915 | Human | pDC | IL3 | Flu | 6h |
| GSE57915 | Human | pDC | GM-CSF | Flu | 6h |
| GSE57915 | Human | pDC | GM-CSF | Flu | 1d |
| GSE57915 | Human | pDC | GM-CSF | LL37 | 1d |
| GSE57915 | Human | Monocytes | NOD2 | TLRs | 6h |
| GSE57915 | Human | Monocytes | NOD2 | TLRs | 1d |
| GSE57915 | Human | Monocytes | IFNγ | TLRs | 6h |
| GSE46903 | Human | Macrophage | IFNγ | TNF | 3d |
| GSE46903 | Human | Macrophage | TNF | P3C | 3d |
| GSE36323 | Human | Monocytic THP-1 | D3 | TsA | 2.5 hr |
| GSE52819 | Human | Macrophage | Vitamin D | H37Rv | 24 hr |
| GSE44392 | Human | CD4+ T cell | edelfosine | beads | 30 hr |
| GSE24767 | Human | Keratinocyte | IL-17 | TNF | 1d |
| GSE77814 | Human | BMSC | IFNγ | TNF (1.5 ng/ml) | 2d |
| GSE77814 | Human | BMSC | IFNγ | TNF (15 ng/ml) | 2d |
| GSE134209 | Human | moDC | TNF | IFNβ | 1h |
| GSE134209 | Human | moDC | TNF | IFNβ | 2.5 hr |
| GSE20302 | Mouse | DC | Lact acidophilus | Bifid bifidum | 10 hr |
| GSE28994 | Mouse | Lung | Pam2CSK4 | ODN2395 | 4h |
| GSE32986 | Mouse | DC | Curdlan (1 mg/ml) | GM-CSF | 4h |
| GSE32986 | Mouse | DC | Curdlan (100 mg/ml) | GM-CSF | 4h |
| GSE35291 | Mouse | HSPCs | Valproic acid | lithium | 7d |
| GSE53986 | Mouse | macrophage | IFNγ | LPS | 1d |
| GSE62249 | Mouse | SB-3123p cells | Cocktail (TNF/IFNγ) | Vemurafenib | 4d |

Non-standard abbreviations: MDM = Monocyte Derived Macrophages; P3C = Pam3 CSK; D3 = nuclear hormone 1,25(OH)2D3; TsA = trichostatin A; BMSC = Bone Marrow Stromal Cells; HSPCs = hematopoietic stem/progenitor cells; Lact = Lactobacillus; Bifid = Bifidobacterium.

For example, slicing by gene family allows systematic identification of positive and negative interactions involving immune modulators of interest. *Figure 2D* shows a 2D projection of the landscape involving stimulatory and inhibitory checkpoints for a selection of datasets including macrophage, moDC, pDC, and T cells. Genes in these families show a variable propensity toward synergistic and antagonistic regulation across datasets. While *CD40* and *CD80* presented sparse, selective interaction effects across datasets, *IDO1* shows positive interactions in 5 out of the 15 selected datasets. The profiles of these synergies contain two cases of potentiation from different experimental models, one involving pDC stimulated with IL3 and influenza virus (Flu), and the other involving moDC stimulated with TNF and IFNβ (*Figure 2E*). Because *IDO1* inhibits T cell division and promotes regulatory T cells (*van Baren and Van den Eynde, 2015*), these positive interactions may serve to restrict overreacting immune responses in the inflammatory microenvironment.

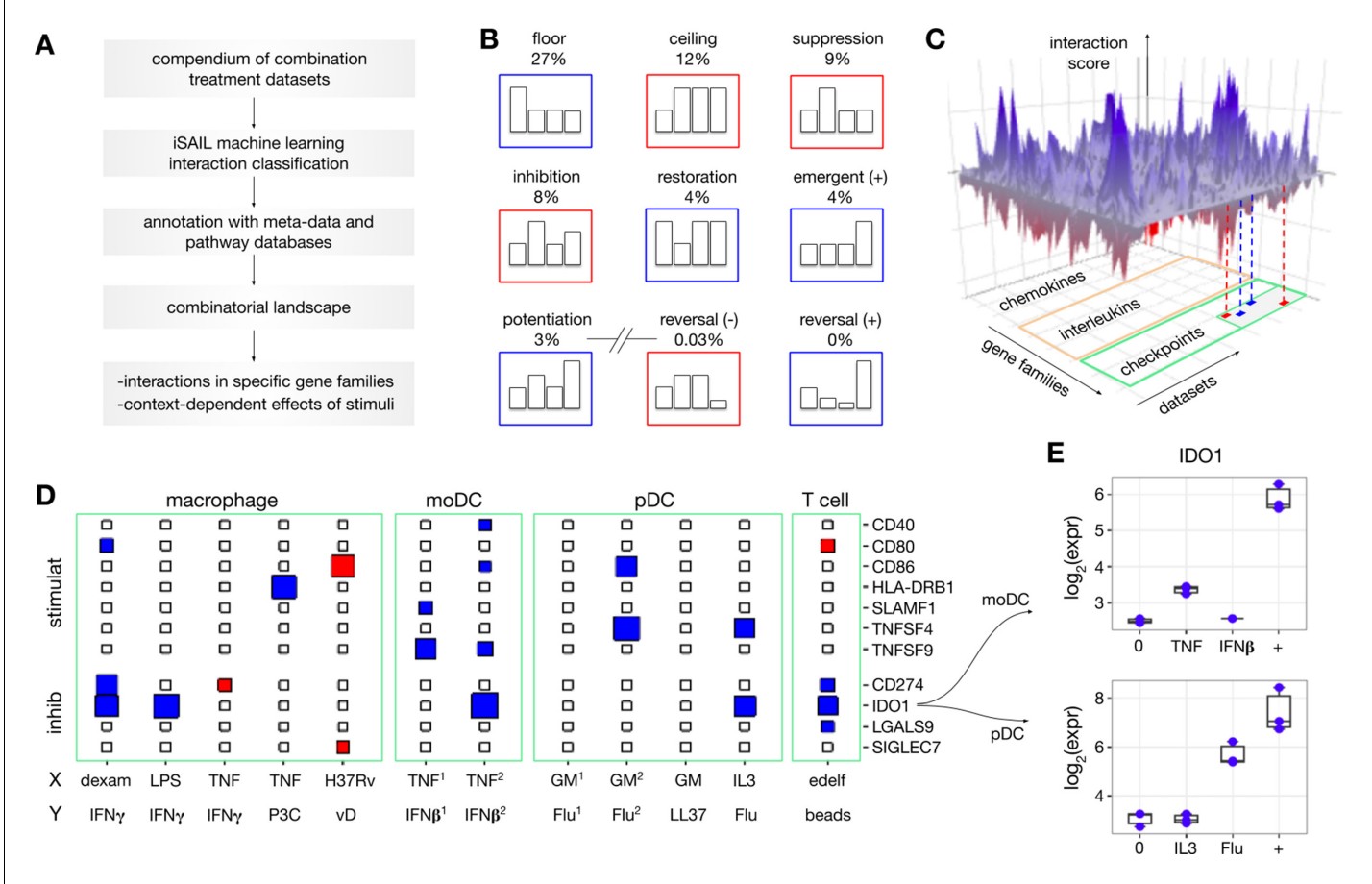

**Figure 2.** Building a combinatorial landscape of immunity. (**A**) We developed a strategy to map and investigate combinatorial effects from multiple combination treatment experiments. Using public repositories, we selected 32 -omics datasets of combination treatment experiments from diverse immune cells and combinations of stimuli. For each dataset, we applied iSAIL and classified a total of ~30,000 interaction effects. (**B**) The first seven cards represent the most frequent interactions across datasets. The last two cards represent interactions that occur with vanishingly low frequency. (**C**) To integrate interactions from different datasets, we created a 3D structure with axes representing datasets, genes, and scores quantifying the intensity and robustness of the effects. The resulting landscape, supplemented with metadata and prior knowledge, makes it possible to comprehensively investigate the effect of combination treatments on immune cells. (**D**) The plane shows a 2D projection of the landscape focusing on immune checkpoints in selected datasets: (stimulat = stimulatory; inhibit = inhibitory). The size and color of the rectangles keep track respectively of the interaction score and sign (blues: positive interactions, red: negative interactions). The immune checkpoint *IDO1* is synergistically induced in multiple datasets. Non-standard abbreviations: P3C = Pam3 CSK; vD = vitamin D; GM = GM CSF; edelf = edelfosine; Flu = influenza virus. (**E**) By looking at the specific nature of these synergies, we found two cases of potentiation in human moDC (top) and pDC (bottom).

The online version of this article includes the following figure supplement(s) for figure 2:

**Figure supplement 1.** Quantification of interaction by cell type and type of combination.

Our results show that mining the interaction landscape can reveal new insight on combinatorial interactions and their immunological functions.

## Interactions determine a context-dependent TNF biology

To further illustrate how the interaction landscape can be analyzed, we study how TNF biology can be altered by the presence of cofactors. To this end, we sliced the combinatorial landscape along two axes (*Figure 3*, top-left). From the dataset axis, we selected combination treatments involving TNF with other stimuli, including IFNγ and IFNβ in four human cell models (*Figure 3*, left-margin). From the interaction axis, we extracted interaction profiles that encoded a qualitative change of the TNF effect when considered as a mono-treatment. We started by considering three types of

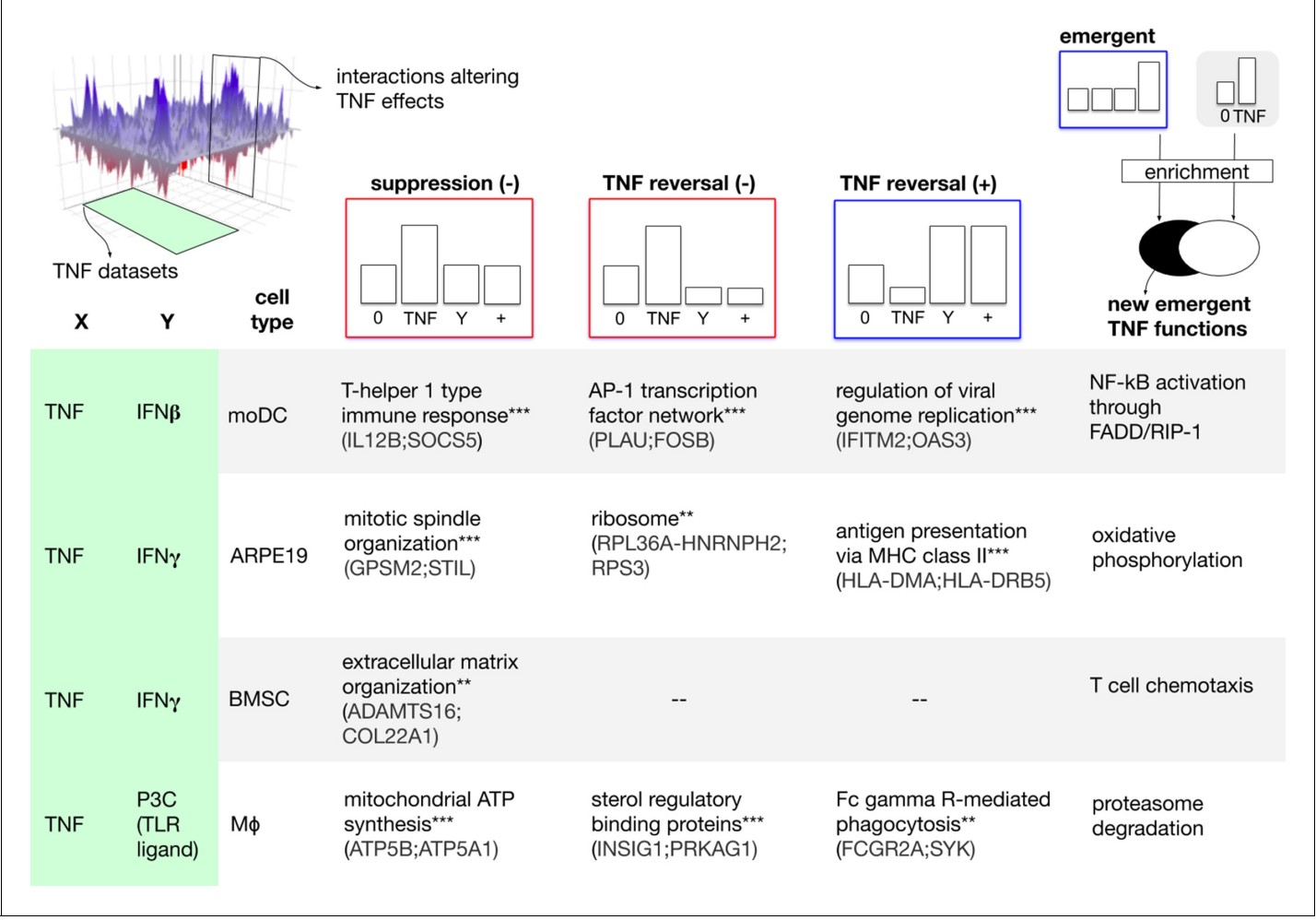

**Figure 3.** Combinatorial interactions determine a context-dependent TNF biology. To explore how cofactors might alter the TNF biology, we sliced the combinatorial landscape (top-left) along two axes. From the dataset axis, we extracted combination treatments involving TNF and concomitant factors including IFNγ and IFNβ in four human cellular models (left-margin). From the interaction axis, we extracted interaction profiles that encode a qualitative change of the effect of TNF mono-treatment. We started by considering three types of qualitative changes: suppression, antagonistic reversal, and synergistic reversal (top-margin). For each dataset and profile, we processed the corresponding gene list with enrichment analysis to gain insight at functional level. The matrix elements correspond to selected significantly enriched functions (hypergeometric test; * adjusted p<0.05, ** adjusted p<0.01, *** adjusted p<0.001). Example hits from each function are shown in parentheses. In the case of emergent effects (last column), we looked for the presence of new functions, not regulated by TNF alone (top right, see also Materials and methods). The results suggest that cofactors could drive the emergence of new TNF functions.

qualitative changes: suppression, antagonistic reversal, and synergistic reversal of TNF effects (*Figure 3*, top-margin).

For each pair of dataset and profile, we processed the corresponding gene list with enrichment analysis to gain insight at the functional level. Genes showing suppression and reversal of TNF effects were significantly enriched in important immune processes including Th1 polarization and antigen presentation (*Figure 3*). Further analysis (*Figure 3*, top-right, Materials and methods) also suggested that co-modulators can drive the emergence of new functions, not observed by TNF stimulation in isolation. Although these emerging functions were relatively few, they comprised key processes such as proteasome degradation, an alternative NF-kB activation pathway, and T cell chemotaxis.

Our results suggest that TNF biology can be qualitatively altered by other stimuli through a variety of interaction effects including suppression, reversal, and the emergence of entirely new functions. This provides a new insight into the function of a cytokine.

## Prediction and validation of TNF and IFNβ interaction effects in human monocyte-derived dendritic cells

Finally, we investigate in-depth the interactions in our newly generated dataset of TNF combined with the cytokine IFNβ (see Materials and methods), and illustrate the hypothesis generation and validation cycle enabled by iSAIL. Although TNF and IFNβ are key modulators of immune function whose individual effects have been extensively studied, their interactions remain poorly understood. We previously reported that TNF and IFNβ act synergistically to induce an antiviral state in monocyte-derived dendritic cells (moDC) (*Hartmann et al., 2014*). To study the systems level impact of TNF and IFNβ co-treatment on human moDC, we applied iSAIL to this transcriptomic combination treatment experiment.

iSAIL detected 374 positive interactions, which we mapped to the corresponding profile groups. The interaction groups with the largest number of positive interactions were 'emergent synergy', 'TNF potentiates IFNβ', 'IFNβ restores TNF', and 'IFNβ potentiates TNF' (*Figure 4A*). To understand the function of these interaction profiles, we sorted the corresponding gene lists based on the synergy score and searched for candidates with potential key immunological roles.

Focusing on emergent synergies, we found the largest synergy scores for *LIMK2*, *MCOLN2*, *SLC7A5*, *TP53BP2*, and several genes having more established immunological roles such as *RELB*, *IL15RA*, and *VCAM1*. The protein VCAM-1 has been described as a regulator of leukocyte migration and cell adhesion (*Deem et al., 2007*). Due to the fundamental importance of the DC-T cell axis in the generation of an immune response, we hypothesized that TNF and IFNβ synergistically induce VCAM-1 to promote moDC-T cell adhesion. We experimentally tested this hypothesis by quantifying DC-T cell adhesion using imaging flow cytometry (see Materials and methods). When exposed to the combination of TNF and IFNβ, moDC showed an increased adhesion to T cells that was not observed with either cytokine alone (*Figure 4B*). The increased DC-T cell adhesion was mediated by VCAM-1, since VCAM-1 neutralization abolished the synergistic effect (*Figure 4B*). To our knowledge, these results identify for the first time a role for synergistic induction of VCAM-1 by TNF and IFNβ in promoting DC-T cell adhesion.

To further explore the immune processes controlled by TNF and IFNβ synergies, we performed enrichment analysis (*Figure 4C*, left) separately for the different profiles, and compared the results with a conventional analysis that aggregates all the synergies in a single gene set (*Figure 4C*, right). Annotation by profile increased the number and significance of annotation terms compared to the conventional analysis. Interestingly, we found a reciprocal potentiation of TNF and IFNβ signaling pathways by the combination treatment. Both iSAIL and conventional analysis captured a highly significant enrichment in mineral absorption. However, iSAIL analysis also revealed the pattern 'IFNβ restores TNF' as the main contributor to this enrichment. This pattern contains several members of the family of metallothioneins (*MT1X*, *MT1E*, *MT1F*, *MT1HL1*), which are increasingly recognized as important players in the response to cytokines and pathogen signals (*Subramanian Vignesh and Deepe Jr., 2017*).

Some annotation terms appeared only in the iSAIL analysis and were not resolved by conventional analysis. In particular, we found the pattern 'TNF potentiates IFNβ' enriched in NK cell proliferation, and the pattern 'IFNβ potentiates TNF' enriched in T cell proliferation. We studied allogeneic cross-donor stimulation to test the hypothesis suggested by this analysis that a synergistic effect on T cell proliferation resulted from IFNβ and TNF stimulation (*Figure 4D*, left panel). We observed a nearly twofold increase in the percent of proliferating T cells upon the combination treatment, an effect not seen with either stimulus alone. The synergy pattern of the T cell proliferation measurement diverged slightly from the gene level profile, which showed some effect by TNF alone. It is not surprising that the interactions profiles comparing mRNA level regulation and protein-dependent functional effects show marginal differences. Importantly, the synergistic induction of T cell proliferation, predicted by iSAIL, was validated experimentally.

Next, we wanted to identify the molecular mediators of the increased T cell proliferation. Candidate genes for mediating this synergistic response predicted by enrichment analysis included *TNSFS9* and *CCL5*. Review of the literature suggested CCL-5 as the most likely candidate (*Makino et al., 2002*). We therefore hypothesized that CCL-5 may contribute to increased proliferation of T cells induced by TNF and INFβ exposed DC. This hypothesis was confirmed by

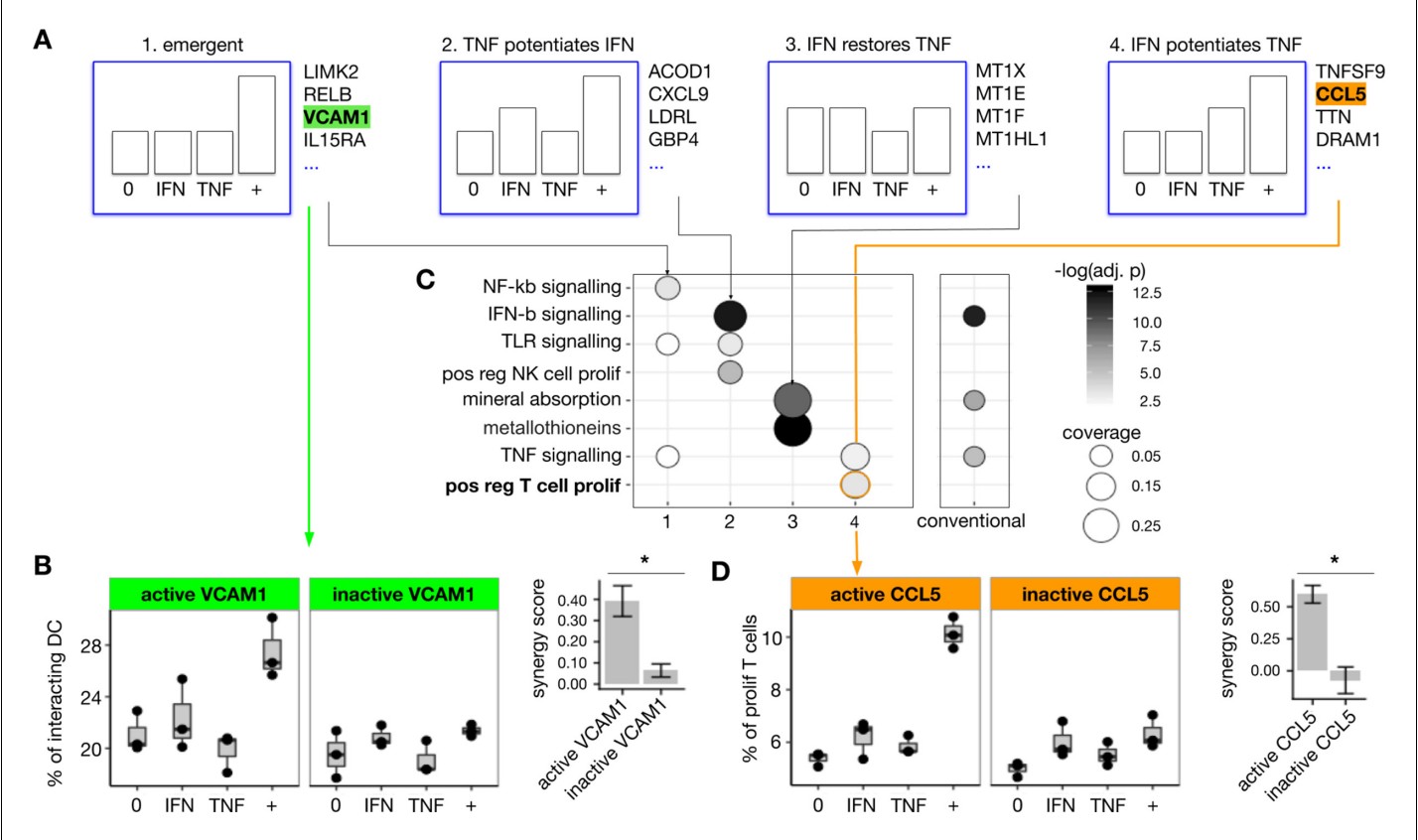

**Figure 4.** Prediction and validation of IFNβ and TNF synergistic effects in monocyte-derived dendritic cells . (**A**) We applied iSAIL to analyze the synergistic effects induced by IFNβ (3000 pg/mL) and TNF (4500 pg/mL) on monocyte-derived dendritic cells after 1 hr of exposure. We focused on the four profile groups with the largest number of synergies. In the group of emergent synergies, we found *VCAM1*, a gene involved in the regulation of cell adhesion. (**B**) We tested the hypothesis that the emergent induction of *VCAM1* in the DC would mediate an increase in DC-T cell interaction. Assayed by imaging flow cytometry, moDCs exposed to IFNβ+TNF showed an emergent increase in DC-T cell adhesion. VCAM-1 neutralization abolished this synergistic effect. The right subpanel shows the mean synergy score of the three replicates, plus or minus the standard error of the mean. VCAM-1 neutralization significantly reduced the synergy score (t-test, p=0.03). (**C**) To explore whether TNF+IFNβ synergy profiles represented coherent gene programs, we determined the functional enrichment for each of the four patterns, shown in the columns. (**C**, left panel). We compared these iSAIL-based functional analyses to results obtained with a conventional analysis of unclassified synergy genes. (**C**, right panel). iSAIL-based enrichment provided a richer functional annotation. In particular, it suggested that synergy genes in the 'IFNβ potentiates TNF' pattern rightmost panel in (**A**) may mediate T cell proliferation. (**D**) This hypothesis was tested using allogeneic cross donor stimulation. The synergy group 'IFNβ potentiates TNF' contained two hits potentially responsible for T cell proliferation: *TNFSF9* and *CCL5*. Using CCL-5 neutralizing antibodies, we confirmed that IFNβ and TNF act in synergy to promote T cell proliferation, and that this proliferation depends on CCL-5. The right subpanel shows the mean synergy score of the three replicates, plus or minus the standard error of the mean. CCL-5 neutralization significantly reduced the synergy score (t-test, p=0.008).

immunoneutralization of CCL-5 (*Figure 4D*, right panel). These validation experiments demonstrate the value of iSAIL in mining interaction data for new hypotheses leading to biological and mechanistic insight.

## Discussion

In this work, we present a comprehensive resource to map and interpret interaction effects from individual and multiple combination treatment studies. Applying our machine learning classification method to the compendium of immunological combination treatment datasets, we identify ~30,000 interactions. We show that the collection of interactions visualized as a landscape can be mined for general insight into the microenvironment-dependent role of key immunomodulators, for specific testable hypotheses about combinatorial control of immune processes.

A potential consequence of interactions is the radical modification of effects observed with individual treatments. Using the iSAIL framework, these events are easily identified by isolating

interaction profiles that encode qualitative combination changes in the effect of a treatment of interest. In the case of TNF, we found that co-modulators alter fundamental immunological processes, such as antigen presentation and T helper cell polarization, and may produce the emergence of entirely new functions, not modulated by TNF exposure alone. Identifying the context-dependent effects of an agent may be useful from a therapeutic perspective. Our approach may assist in the design of drug combinations that leverage antagonistic interactions to selectively reduce pathogenic activity while preserving necessary homeostatic effects.

A fundamental tool for biological interpretation of -omics experiments is pathway-level enrichment analysis. iSAIL uncovers biological processes regulated in combination treatment studies by fine-grained aggregation of genes showing similar interaction responses. These aggregates can reflect coherent biological processes, providing insight into the principles and functions of interactions. In the analysis of TNF and IFNβ co-treatment, iSAIL uncovered novel synergies that control the DC-T cell interactions and T cell proliferation, both of which are critical immune processes. Importantly, iSAIL analysis generated hypotheses on the molecular mediators controlling these functions, VCAM-1 and CCL-5, which were validated by additional experiments. The ability to identify new interactions and mechanisms in a relatively well characterized combination such as TNF and IFNβ points to an even greater potential for discovery in less characterized combinations.

In its current version, iSAIL was developed to analyze transcriptomics experiments. Consistent with this, the classifier was trained under the assumption of normally distributed data, which is commonly held in the analysis of microarray and RNA-seq data (upon a suitable transformation). However, under alternative distributional assumptions, our framework can be adapted to any -omics data from combination treatment experiments, including proteomics, metabolomics, and epigenomics.

Understanding the dynamic properties of combinatorial interactions is a very relevant, yet largely unexplored area. Included in our compendium are time-course datasets with only two time points. Due to such poor temporal sampling, these studies were analyzed as separate experiments. Although the dynamics of combinatorial responses is not addressed in our work, the iSAIL framework would still provide a fundamental grammar to map gene trajectories from longer time series. Time-dependent interrelationships could be included in our analysis, for example, by imposing additional constraints that guarantee smooth transitions between interaction profiles at successive time points. Our analysis framework, applied to a large compendium of time course experiments, should they become available, will enable the identification of preferential trajectories and dynamic rules governing time transitions between interaction profiles.

Despite the centrality of combinatorial interactions to the control of the immune system, many important questions remain open. In particular, the ability to predict combinatorial interaction effects based on the individual effects remains elusive (*Preuer et al., 2018*; *Ryall and Tan, 2015*; *Menden et al., 2019*). Although predicting interactions is unrelated to the objectives of our study, iSAIL may still support research in this direction. For example, our global estimates of the frequency of different interactions may be used as prior information to enhance the predictive power of future models.

Finally, insight into combination treatment experiments is relevant to the field of drug therapeutics, as thousands of clinical trials in the United States alone are studying the effects of drug combinations (*Schmidt, 2017*; *Nature Medicine, 2017*; *Wu et al., 2015*). Drug interactions can drive both clinical therapeutic efficacy and treatment side effects. The success of drug combinations relies on the control of pathogenic pathways while preserving the homeostatic pathways. By uncovering relevant interactions and their functions, iSAIL user-friendly online tools can further the understanding and therapeutic applications of interaction mechanisms.

## Materials and methods

### Definition and simulation of interaction profiles

The notion of interaction profile was introduced in our previous work (*Cappuccio et al., 2015*) and is briefly summarized here. *Figure 1—figure supplement 1* shows the taxonomy of 123 profiles used in this study. Mathematically, each of these profiles corresponds to a linear system of inequalities satisfied by the mean expression levels of a gene in the conditions 0, X, Y, X+Y. These mean expression levels are respectively denoted by $\overline{e_O}$, $\overline{e_X}$, $\overline{e_Y}$, $\overline{e_{X+Y}}$. The system that defines a given profile

admits infinitely many solutions, each of which can be seen as a particular instance of the profile. For example, an emergent synergy (*Figure 1—figure supplement 1*, profile 3) is satisfied by the vectors (2.2, 2.2, 2.2, 5.5), (4.1, 4.1, 4.1, 7.8), as well as by infinitely many other qualitatively similar vectors.

To simulate a profile, our strategy starts by sampling the solution space of the corresponding system of inequalities. The solution space is sampled within a range of admissible values, calibrated to mimic the typical numerical ranges of -*omics* data. Next, a noise term is added to each instance of a profile using a random number generator. To account for the relatively small number of samples in -*omics* data, we simulated four replicates for each of the conditions 0, X, Y, X+Y. The noise term is assumed to be normally distributed. This assumption is widely held in the analysis of microarray data, and still applicable to RNA-seq data upon a suitable transformation (*Law et al., 2014*). Increasing noise levels correspond to a decreasing effect size, which is defined in terms of the standardized differences between the group means in the four conditions, as further described below.

The steps to simulate interaction profiles are as follows:

*Definition of a range of admissible expression values.* This was chosen as the interval $[-14, 14]$, consistent with the log2-transformed expression values from microarray and RNA-seq upon a Voom transformation (*Law et al., 2014*).

*Sampling the solution space for the given profile in the specified range.* The vector of numbers $(\overline{e_0}, \overline{e_X}, \overline{e_Y}, \overline{e_{X+Y}})$ for the given profile were found with the function *xsample* from the package *limsolve*. For each profile, we extracted 400 instances for each level of noise.

*Definition of the signal of a simulated profile.* The signal can be seen as a generalization of the fold-change in A vs. B experiments. In a combination treatment, we first consider all pairwise fold-changes from the conditions 0, X, Y, X+Y. Except for the case of constant genes, at least one of these contrasts must be different from 0. The signal $\delta$ is defined as the absolute value of the smallest non-zero differences. To avoid very weak signals, not meaningful in the analysis of expression data, a minimum signal of 0.5 is used in the simulated data.

*Simulation of random noise.* To simulate random variability around the values $\overline{e_0}, \overline{e_X}, \overline{e_Y}, \overline{e_{X+Y}}$, the data was assumed to be normally distributed around the group means: $e_i \sim N(\overline{e_i}, \sigma)$, with $i = 0, X, Y, X+Y$. The parameter $\sigma$ was assumed the same for all groups. For each of the four conditions, we simulated $n=4$ replicates. Different levels of noise, were simulated by setting different values of the ratio $\delta/\sigma$.

*Enforcement of the range of the expression values.* The addition of random noise can push some of the values $e_i$ outside the initially prescribed range of expression. In this case, we forced the simulated values to be at the limit of the range. For example, a value of -18.5 was reset to -14, the lower limit of the prescribed range.

## Training and testing of the machine learning classifier

To train the machine learning classifiers, we generated a training set by simulating multiple instances of every profile in the admissible range of expression values. Each instance of a simulated profile was represented as a vector of statistical features. These include the estimated mean values $\overline{e_0}, \overline{e_X}, \overline{e_Y}, \overline{e_{X+Y}}$, the p-values of all possible pairwise contrasts among these four values, and additional statistics returned by the *Limma* package (*Ritchie et al., 2015*) served as predictors of the true class.

The training set comprised different noise regimes. These were simulated by fixing different values for the parameter $\delta/\sigma$, as described in the previous section. We considered the following values: low noise ($\delta/\sigma = 4$), medium noise ($\delta/\sigma = 2.5$), high noise ($\delta/\sigma = 2$). Training of the machine learning classifiers was done using the R packages *Caret* and *RandomForest*.

To select the best model, we generated additional simulated data and measured the out-of-sample classification accuracy per profile and for the same values of $\delta/\sigma$ that were used to build the training set. For each of these values, the accuracy was quantified as the proportion of correct predictions. An advantage of RF and LDA over the deterministic match is the possibility for a 'soft' (i.e. probabilistic) classification of an input profile into any element of the taxonomy. The quality of the full probabilistic output of RF and LDA was quantified for all noise levels using the multi-class log gain (*Figure 1—figure supplement 2D*). To further assess the performance of RF and LDA for the different classes, the distribution of precision and recall over all taxonomy classes was examined (*Figure 1—figure supplement 2*). All these criteria indicated RF as the most robust classifier. The multiclass log-gain, and the class-specific precision and recall were computed with the R packages *MLmetrics* and *mltest*.

## Generating the combinatorial landscape of immunity

Public combination treatment datasets were retrieved from Gene Expression Omnibus using the package *GEOquery* (*Davis and Meltzer, 2007*). The relevant datasets were identified using key terms typically associated with combination treatments such as 'synergy', 'antagonism', 'combinatorial', and similar. This approach was designed to facilitate automatic update of the resource as new -*omics* data from combination treatment experiments become publicly available. To facilitate comparisons, all the datasets were imported in the format of the original publications. The datasets were preprocessed as follows. First, if different probes were available for the same gene, the probe with the largest coefficient of variation was selected. Second, genes with low coefficient of variation (lower than the median across all genes) were filtered out. Next, differentially expressed genes were determined with the Limma package. A significance cutoff of 0.05 was applied on the p-values corrected for multiple testing. An additional cutoff was imposed on the δ (defined above): genes with δ lower than the median value computed across all differentially expressed genes were filtered out. The resulting differentially expressed genes were then analyzed by the machine learning classifier which assigned to each gene the predicted element of the taxonomy. For each identified interaction, a score was defined to measure its magnitude as well as its significance. The magnitude of the interaction $b$ was measured as the average Bliss index, defined as the average deviation from additivity $b = \Delta\overline{e_{X+Y}} - (\Delta\overline{e_X} + \Delta\overline{e_Y})$, where $\Delta\overline{e_i} = \overline{e_i} - \overline{e_0}$ ($i$ = X, Y, X+Y), $b>0$ for positive interactions and $b<0$ for negative interactions. The significance of the interaction was measured as the class probability $p$ returned by the classifier. To account for both the significance and the magnitude of an interaction, an overall score was defined as the product $b \cdot p$.

The identified interactions were annotated using a manually curated list of stimulatory and inhibitory immune checkpoints, as well as the gene lists provided by the ImmPort database (*Bhattacharya et al., 2014*).

## Enrichment analysis

The functional enrichment of interactions was done using the Enrichr library (*Kuleshov et al., 2016*). Four annotation databases were considered: GO Biological Processes (2017b), KEGG (2016), Wikipathways (2016), Reactome (2016). Enrichment was considered significant if the enrichment p-value adjusted for multiple testing was lower than 0.05.

To analyze the synergies induced by IFNβ and TNF co-treatment, we focused on annotation terms with size lower than 500 genes, to increase the specificity of the identified functions and pathways. Moreover, we imposed a minimum threshold on the overlap between the annotation term and the gene set being analyzed. This threshold was meant to identify annotation terms covering a minimum proportion of the gene set being analyzed. We chose a minimum coverage of 2%.

## DC differentiation

All human subjects research protocols were reviewed and approved by the IRB of the Icahn School of Medicine at Mount Sinai. Monocyte-derived DCs were obtained from healthy human blood donors following a standard protocol described elsewhere (*Hartmann et al., 2017*). All experiments were replicated using cells obtained from different donors.

## Microarray data of human moDC treated with TNF and IFNβ

DC were treated with 4500 pg/mL TNF, 3000 pg/mL IFNβ, or the combination of both for either 1 hr or 2.5 hr. Untreated DC served as a negative control. Three samples were taken per treatment and time point. RNA was extracted with the RNeasy plus kit (Qiagen) following the manufacturer's instructions. Gene expression was assayed using broad human genome specific HG-U133_Plus_2 GeneChip expression probe arrays (Affymetrix). Affymetrix microarray data were normalized using gcRMA (*Wu and Irizarry, 2004*). Additional data processing was done as described above (see in Materials and methods section 'Generating the combinatorial landscape of immunity').

## Experimental validation of TNF and IFNβ synergistic effects

To test the involvement of VCAM-1 in mediating TNF and IFNβ induced synergy, DCs were exposed to TNF, IFNβ, the combination of TNF and IFNβ or control as described above. Four hours after treatment DCs were exposed to allogeneic T cells in a 1:3 ratio for an additional 4 hr and then fixed

with paraformaldehyde. Cells were stained with fluorochrome labeled antibodies against CD11c (DCs) and CD3 (T cells) and analyzed by imaging flow cytometry. DCs interacting with T cells were identified in images where one or multiple T cells had a direct contact with a DC.

To test the involvement of CCL-5 on TNF and IFNβ induced synergy, DCs were exposed to the cytokine mixtures as described above. After 4 hr, DCs were exposed to CFSE stained allogeneic T cells for 5 days and then fixed with paraformaldehyde. Cells were stained with a monoclonal antibody against CD11c and the extent of T cell proliferation was measured by the dilution of CFSE in the CD11c-negative population, as CFSE gets weaker with every T cell division. The stain to exclude DCs was necessary as DCs also digest CFSE-positive parts of T cells.

## Acknowledgements

This work was supported by the US National Institutes of Health (NIH) grant 5U19AI117873, by the Defense Advanced Research Projects Agency (DARPA) contract number N6600119C4022, by the European Research Council under Grant IT-DC 281987, by Agence Nationale de la Recherche under Grant ANR-11-LABX-0043 CIC IGR-Curie 1428, and by ERC 2015 POC DrugSynergy 680890.

## Additional information

### Funding

| Funder | Grant reference number | Author |
|---|---|---|
| National Institute of Allergy and Infectious Diseases | 5U19AI117873 | Stuart C Sealfon |
| Defense Advanced Research Projects Agency | N6600119C4022 | Stuart C Sealfon |
| European Research Council | IT-DC 281987 | Vassili Soumelis |
| Agence Nationale de la Recherche | ANR-11-LABX-0043 CIC IGR-Curie 1428 | Vassili Soumelis |
| European Research Council | POC DrugSynergy 680890 | Vassili Soumelis |

The funders had no role in study design, data collection and interpretation, or the decision to submit the work for publication.

### Author contributions

Antonio Cappuccio, Conceptualization, Resources, Data curation, Software, Formal analysis, Supervision, Funding acquisition, Visualization, Methodology, Writing - original draft, Project administration, Writing - review and editing; Shane T Jensen, Formal analysis, Methodology, Writing - review and editing; Boris M Hartmann, Conceptualization, Supervision, Funding acquisition, Validation, Methodology, Writing - original draft, Project administration, Writing - review and editing; Stuart C Sealfon, Conceptualization, Formal analysis, Supervision, Funding acquisition, Writing - original draft, Project administration, Writing - review and editing; Vassili Soumelis, Conceptualization, Supervision, Funding acquisition, Writing - review and editing; Elena Zaslavsky, Conceptualization, Formal analysis, Supervision, Funding acquisition, Validation, Methodology, Writing - original draft, Project administration, Writing - review and editing

### Author ORCIDs

Boris M Hartmann http://orcid.org/0000-0002-5649-6776
Elena Zaslavsky https://orcid.org/0000-0002-4828-7771

### Decision letter and Author response

Decision letter https://doi.org/10.7554/eLife.62148.sa1
Author response https://doi.org/10.7554/eLife.62148.sa2

## Additional files

### Supplementary files

- Source data 1. VCAM-1 neutralizing experiment.
- Source data 2. CCL-5 neutralizing experiment.
- Transparent reporting form

### Data availability

Gene expression data generated for this study have been deposited in GEO under accession code GSE134209. All other dataset accession codes analysed in this study are included in the manuscript (Table 1). All the analyses have been implemented in R. An interactive R Shiny application of iSAIL can be found at https://isail.shinyapps.io/test_app/. The site also contains downloadable code and documentation to run the software locally.

The following dataset was generated:

| Author(s) | Year | Dataset title | Dataset URL | Database and Identifier |
|---|---|---|---|---|
| Hartmann BM, Zaslavsky E | 2019 | Microarray data of human moDC treated with IFN$\beta$ and TNF$\alpha$ | https://www.ncbi.nlm.nih.gov/geo/query/acc.cgi?acc=GSE134209 | NCBI Gene Expression Omnibus, GSE134209 |

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
