## [Decision Letter]

**Acceptance summary:**

This resource aggregates information on interactions from public immunological

(interleukins, checkpoints, and other immune modulators) data sets to facilitate the analysis of combinations of drug and ligands onto immune cell activation. The system, based on underlying random-forest classification to sort out experimental noise and reveal new properties, provides information at the level of general functional classes (enrichment analysis) associated to estimated error levels, within a user oriented graphical interface.

**Decision letter after peer review:**

[Editors’ note: the authors submitted for reconsideration following the decision after peer review. What follows is the decision letter after the first round of review.]

Thank you for submitting your work entitled "Deciphering the combinatorial landscape of immunity" for consideration by *eLife*. Your article has been reviewed by three peer reviewers, one of whom is a member of our Board of Reviewing Editors, and the evaluation has been overseen by a Senior Editor. The reviewers have opted to remain anonymous.

Our decision has been reached after consultation between the reviewers. Based on these discussions and the individual reviews below, we regret to inform you that your work will not be considered further for publication in *eLife*.

The paper describes an method to study interactions in a large immunological molecular system of ~30,000 interactions: The main hypothesis derived from the system is related with the immunological effects of TNFα on IFNβ.

There are a number of problems with the current submission. First, the proposed method has not been systematically validated beyond the arguments in this paper. The example chosen corresponds to a well-known system, e.g. TNF cytokines synergies, for which it is unclear how much the paper adds. In any case, to have a significant impact in biology a more challenging hypothesis , going beyond the already-known synergies of TNF, should be proposed and ideally, it would be followed by additional experiments. Only then the potential of the system will be demonstrated as actually able to uncover new biology.

While based on this criticism the paper will not be acceptable for publication, it might be considered if organized as a resource paper (A Tools and Resources article allows authors to publish the details of new experimental techniques, datasets, software tools, and other resources). It will be also possible to consider a future version of this work including a complementary validation of the method and experimental follow-up of the proposed models.

Reviewer #1:

The authors have developed an interesting resource with 30k interactions extracted from public immunological datasets (25 human and 7 murine combination treatment studies).

On this data set they apply a aggrupation and training strategy (random forest algorithm operating on what are called "logic principles" of the interactions) to differentiate single responses from combine ones (synergies) in a set of interleukins, checkpoints, and other immune modulators. The system works at a semi-qualitative level that takes into account the fluctuations and variability of the data, and provides an interpretation at the level of general functional classes (enrichment analysis).

The potentially more interesting innovation of the model is the systematic introduction of error levels. The most valuable asset is the collection of interactions from public data. The proposed machine learning technology, the biological analysis and the grouping of genes is also rather standard and simple. The visualization is nice and help to understand the results.

The key problem is that the method is not validated in any systematic way. The proposed experimental follow-up might be interesting from a biological point of view but cannot be considered systematic validation. It might be a nice tool for exploration but the information presented is not enough to consider it as a prediction system.

Other points are related with the way the methodology is presented, that is rather obscure when in practice it is very simple, i.e. functional inference is enrichment analysis, machine learning interaction classification is a random forest classifier, omic analysis is expression data, etc. While the main idea of combining and averaging results is not explained clearly.

My opinion is that this paper could be considered for publication if it can be presented as a resource for the exploration of immunological interactions based on the collected data, gene aggregation strategy and visualisation system. In this case, it would have to be written in a different way, clearly oriented to use of the system and providing the biological results as examples of use.

Reviewer #2:

In this article, Cappuccio et al. introduce a computational method to better assess the impact of combinations of drug and ligands onto immune cell activation. This is a problem of fundamental significance, both from the theoretical side (how to tackle the combinatorial complexity of immunological interactions?) and from the practical side (how to leverage this combinatorial complexity to maximize clinical perturbation?). There exist many experimental datasets that have attempted at reporting the transcriptional responses of cells under single and dual perturbations, yet, there does not exist a systematic framework to integrate, classify and leverage these datasets. This publication introduces a Synergistic/Antagonistic Interaction Learner (iSAIL) to tackle this issue and to publicize the method.

One of the key premises of this computational framework is that individual measurement can lead to inaccurate classification of molecular perturbations, because of limited signal/noise and/or saturation. However, when leveraging the large number of datasets already available, one can improve on this computational task and get a more accurate classification. Hence, the key novelty in this manuscript is presented in Figure 2 and 3 where the statistical framework is introduced. Again, the sobering experimental fact is that individual experiments (e.g. dataset acquired for a given cell type and/or a limited set of perturbation) can be misleading classifying how two molecular perturbations can combine. The computational framework proposed here use random-forest classification to sort out this experimental noise and reveal new properties.

Figure 4 and 5 present one example related to responses to IFN-b and TNF to highlight an emergent function of these cytokines as highly synergistic depending on the molecular/cellular context. One could question whether this method scales up to higher level of combinations of perturbations: given the exponential explosion when combining N drugs, how do the computational framework introduced here help in designing higher order of immunological perturbation?

Overall, this is a serious/robust study addressing an important issue in pharmacology and immunology. Additional details about the method would help the reader better grasp the novelty of the computational method but release of the iSAIL method on a website is making this criticism less stringent: experimentalists will be able to test the method directly and assess its usefulness. More specific comments/suggestions are listed below.

Detailed comments:

Results first paragraph: What is defined here as synergy and antagonism is not congruent with the classical definitions of these terms. One example of this is profile 4 in Figure 1—figure supplement 1, which shows that individual treatments inhibit expression of a particular gene, and that the combination of treatments enhances that inhibition. Classically, this would be considered synergism in the ability of inhibiting expression of that gene, but in your classification this is considered as antagonism, because the final expression levels are lower than the expected by additivity. In your previous paper (Cappuccio et al., 2015), your classification was in agreement with previous definitions, and this interaction was classified as "negative synergy". Unless you provide support and clear explanation for changing this nomenclature, I would recommend going back to the nomenclature used in Cappuccio et al., 2015, namely calling them "positive and negative interactions", rather than "synergistic and antagonistic interactions". Additionally, for cases were the combination interactions were not as dramatic as the few examples showed in the figures, or for profiles called "ceiling" or "floor", it is not possible to determine synergism or antagonism in the absence of dose-response curves obtained by -omics experiments (Chou et al., 2006). The absence of dose-response information would also be problematic for genes which regulation is not monotonic as a function of dose (Senthivel et al., 2016).

Referring to "emergent responses (see Figure 1—figure supplement 1 , profile 3)", make effect bigger in Figure 1—figure supplement 1 so it is clearer to see the differences between profile 3 and 7.

Referring to "would be aggregated with nominal synergistic effects due to frequent cellular or assay saturation, which have low biological significance (see Figure 1—figure supplement 1 , profile 7)”. Please explain better the text before the comma. I think in profile 7 you are referring to the "floor" profile, where background in the assay or basal levels of gene expression makes it look like synergistic, and in your current explanation you are referring to the "ceiling" profile, where saturation is the problem. You can write something similar as it appeared later in the paper regarding these interactions that are likely not meaningful: "represent range limitations of the assay or biological responses"

Multiple instances:

Please replace TNF-α with simply TNF. The former term is obsolete (there is no TNF-β…) and has been retired.

Figure 1: The panels for gene 2 and gene N could be permuted to maintain the hierarchy shown in the second column (synergistic > additive > antagonistic). The graphical rendition of drug interactions as slanted panel in the second column is hard to interpret: what do the authors want to highlight there?

Figure 2D: stimulat -> stimulate

Figure 2E: Specify which IFN.

Figure 5 legend: Include concentrations of IFN-β and TNF (these could be normalized to EC50 to make it easier to read and interpret -is the experiment done at saturation -ceiling-, at subdetectable levels -floor- or in linear regime ;around EC50)

Label the columns in panel c

Reviewer #3:

This study utilizes machine learning to examine gene expression datasets of immune stimulation and attempts to derive a universally applicable set of canonical responses. The basic approach dates back to the work of Janes et al. (Science 2005), which, incidentally, is not cited. My main concern centers on the need for a relatively large amount of data combined with synthetic data in order to obtain insights focused on well-studied interactions of TNF with other cytokines. Furthermore, I am unclear as to the true universal applicability of this approach, for the following reasons:

1) The authors mention that the datasets used involved multiple time points, but it is unclear how dynamics are being handled and modeled across these datasets. One would fully expect that in any extensive time course study of two stimuli, multiple outcomes (e.g. potentiation followed by tolerance or stimulation followed by suppression) would be observed. How can the algorithm then decompose such phenomena into a single characteristic response (i.e., does the algorithm treat each time point as a separate experiment, thereby losing true time-dependent interrelationships?)

2) The system chosen for validation, namely TNF + IFN-β, does not seem particularly novel. It has been appreciated for decades that TNF synergizes with other cytokines to produce an array of effects, including potentiation and tolerance as examples of preconditioning. A demonstration of a truly unexpected set of interactions between stimuli would have been more convincing.

3) The additional validation studies focused on VCAM1 also seem to be focused on relatively well-studied phenomena. Multiple other gene transcripts that appear more novel are not followed up.

---

## [Author Response]

[Editors’ note: the authors resubmitted a revised version of the paper for consideration. What follows is the authors’ response to the first round of review.]

Reviewer #1:The authors have developed an interesting resource with 30k interactions extracted from public immunological datasets ( 25 human and 7 murine combination treatment studies).On this data set they apply a aggrupation and training strategy (random forest algorithm operating on what are called "logic principles" of the interactions) to differentiate single responses from combine ones (synergies) in a set of interleukins, checkpoints, and other immune modulators. The system works at a semi-qualitative level that takes into account the fluctuations and variability of the data, and provides an interpretation at the level of general functional classes (enrichment analysis).The potentially more interesting innovation of the model is the systematic introduction of error levels. The most valuable asset is the collection of interactions from public data. The proposed machine learning technology, the biological analysis and the grouping of genes is also rather standard and simple. The visualization is nice and help to understand the results.The key problem is that the method is not validated in any systematic way. The proposed experimental follow-up might be interesting from a biological point of view but cannot be considered systematic validation. It might be a nice tool for exploration but the information presented is not enough to consider it as a prediction system.

When analyzing combinatorial treatments datasets, there are two major challenges. One is classifying and interpreting the presence of interactions. The second is predicting the occurrence of interactions from single treatment data. From this reviewer’s critique, we suspect that these two different goals (classifying and interpreting vs predicting interactions) have not been clearly distinguished in introducing the problem setting, and this may have caused a lack of clarity regarding the main contributions of our work. In this context, we clarify both the objective and validation of our framework below.

Objective of the method: Our method is not a “prediction system” i.e. our objective is not to predict, denovo, the occurrence of interactions from individual responses. The goal of our method is to identify, using a statistically rigorous approach, to classify and to interpret interactions. Rigorous identification of interaction effects is not a trivial matter. When placed in context, the value of this advance, which is recognized by this reviewer, becomes evident, the rigorous interaction identification component of our work is equivalent to how the study of individual gene fold-changes in transcriptomics datasets only advanced after the systematic introduction of error levels.

Systematic Validation: Consistent with our main objectives of detecting and interpreting interactions, we provide validation of our framework at two levels: (1) by testing the performance of our method on simulated data, where the classification of interactions can be assessed in the presence of the ground truth, and (2) through validation experiments confirming the correct biological interpretation of new synergies between TNF and IFNb.

Finally, while we do not aim to directly predict interactions, we show that our framework does provide new insight on general rules that govern combinatorial interactions. As we state in the discussion, these general rules can be used in future work aiming to predict interactions, but this possibility is tangential to the main research objective of the present manuscript. We acknowledge that our results on the probabilistic algebra may have generated confusion, as that section is thematically close to predictive modelling. Thus, in this revised, resource-oriented version, we have removed this section from the Results.

To clarify these points, we have extensively revised the Introduction and the Discussion sections.

Other points are related with the way the methodology is presented, that is rather obscure when in practice it is very simple, i.e. functional inference is enrichment analysis, machine learning interaction classification is a random forest classifier, omic analysis is expression data, etc. While the main idea of combining and averaging results is not explained clearly.

We appreciate the reviewer’s suggestions and have revised the manuscript to streamline the description of the method and improve the presentation of aggregating gene responses for more fine-grained enrichment analysis.

My opinion is that this paper could be considered for publication if it can be presented as a resource for the exploration of immunological interactions based on the collected data, gene aggregation strategy and visualisation system. In this case, it would have to be written in a different way, clearly oriented to use of the system and providing the biological results as examples of use.

We appreciate the reviewer’s suggestion and are now submitting a revised version of our manuscript for consideration as a Tools and Resources article. In this revised version we present our work as a collection of tools and datasets for the exploration of immunological interactions. The biological results are now presented as a set of use-cases of our system.

Reviewer #2:In this article, Cappuccio et al. introduce a computational method to better assess the impact of combinations of drug and ligands onto immune cell activation. This is a problem of fundamental significance, both from the theoretical side (how to tackle the combinatorial complexity of immunological interactions?) and from the practical side (how to leverage this combinatorial complexity to maximize clinical perturbation?). There exist many experimental datasets that have attempted at reporting the transcriptional responses of cells under single and dual perturbations, yet, there does not exist a systematic framework to integrate, classify and leverage these datasets. This publication introduces a Synergistic/Antagonistic Interaction Learner (iSAIL) to tackle this issue and to publicize the method.One of the key premises of this computational framework is that individual measurement can lead to inaccurate classification of molecular perturbations, because of limited signal/noise and/or saturation. However, when leveraging the large number of datasets already available, one can improve on this computational task and get a more accurate classification. Hence, the key novelty in this manuscript is presented in Figure 2 and 3 where the statistical framework is introduced. Again, the sobering experimental fact is that individual experiments (e.g. dataset acquired for a given cell type and/or a limited set of perturbation) can be misleading classifying how two molecular perturbations can combine. The computational framework proposed here use random-forest classification to sort out this experimental noise and reveal new properties.Figure 4 and 5 present one example related to responses to IFN-b and TNF to highlight an emergent function of these cytokines as highly synergistic depending on the molecular/cellular context. One could question whether this method scales up to higher level of combinations of perturbations: given the exponential explosion when combining N drugs, how do the computational framework introduced here help in designing higher order of immunological perturbation?Overall, this is a serious/robust study addressing an important issue in pharmacology and immunology. Additional details about the method would help the reader better grasp the novelty of the computational method but release of the iSAIL method on a website is making this criticism less stringent: experimentalists will be able to test the method directly and assess its usefulness. More specific comments/suggestions are listed below.

We appreciate the reviewer’s positive evaluation of our study. The reviewer notes both the key methodological novelty of our framework, as well as the extensive validation studies.

Regarding scaling up iSAIL to combinations of more than two stimuli, we would like to emphasize the fact that our approach is not intrinsically limited to pairwise combinations. In principle, the main steps of our framework (e.g., combinatorial analysis, generation of simulated data, and training of a classifier) could be implemented to handle the case of more than two stimuli. However, the main limitation is due to data scarcity. Our search of public data repositories (datasets deposited before 2019) showed that immunological combinatorial datasets with more than two stimuli are extremely rare, precluding systematic research in this direction. Although higher order perturbation combinations are an important aspect of research into combinatorial treatments, due to a complete lack of appropriately structured datasets and to the need for further extensive development, we consider it beyond the current scope of our work. To acknowledge the relevance of the higher order combinations, we have now addressed this point in the Discussion.

Detailed comments:Results first paragraph: What is defined here as synergy and antagonism is not congruent with the classical definitions of these terms. One example of this is profile 4 in Figure 1—figure supplement 1, which shows that individual treatments inhibit expression of a particular gene, and that the combination of treatments enhances that inhibition. Classically, this would be considered synergism in the ability of inhibiting expression of that gene, but in your classification this is considered as antagonism, because the final expression levels are lower than the expected by additivity. In your previous paper (Cappuccio et al., 2015), your classification was in agreement with previous definitions, and this interaction was classified as "negative synergy". Unless you provide support and clear explanation for changing this nomenclature, I would recommend going back to the nomenclature used in Cappuccio et al., 2015, namely calling them "positive and negative interactions", rather than "synergistic and antagonistic interactions". Additionally, for cases were the combination interactions were not as dramatic as the few examples showed in the figures, or for profiles called "ceiling" or "floor", it is not possible to determine synergism or antagonism in the absence of dose-response curves obtained by -omics experiments (Chou et al., 2006). The absence of dose-response information would also be problematic for genes which regulation is not monotonic as a function of dose (Senthivel et al., 2016).

As pointed out by the reviewer, we modified the nomenclature presented in our previous work. One of the main changes, in particular, was to replace the terms of “positive” and “negative” interactions with the terms “synergistic” and “antagonistic” interactions, respectively. However, we would like to point out that, while the nomenclature has been revised, the precise definition of “synergistic” (“positive”) interactions and of “antagonistic” (“negative”) interactions are consistently applied in our previous and current works. In both cases, “synergistic” (positive) interactions comprise super-additive effects, while “antagonistic” (negative) interactions are sub-additive effects. To illustrate this point, we refer to the profiles 4, and 6 in Figure 1—figure supplement 1. From the qualitative point of view, the two profiles represent the same response pattern of a gene in the condition 0, X, Y, X+Y. That is, both profile 4 and 6 express the same ranking of the expression values in the four conditions. Such ranking is compatible both with a positive (synergistic) and with a negative (antagonistic) interaction. However, what makes profile 4 and 6 different is the comparison between the expression value of X+Y and the expected level of additivity. In profile 4 (antagonistic, or negative interaction), the expression value is lower than the expected additivity; in profile 6 (synergistic, or positive interaction), the expression value is larger than the expected additivity.

Due to the reviewer’s comment, we acknowledge that the change of the terms “positive” and “negative” interactions with “synergistic” and “antagonistic” interactions may generate confusion. To address this issue, we accept the reviewer’s suggestion to revert back to our original nomenclature, namely calling "positive and negative interactions", rather than "synergistic and antagonistic interactions".

Regarding the impossibility to determine synergism or antagonism in the absence of dose-response curves obtained, we agree with the reviewer. These effects are problematic and difficult to interpret but, at the same time, very frequent. Unless otherwise specified by the user, iSAIL automatically excludes these ceiling and floor effects from downstream analysis.

Referring to "emergent responses (see Figure 1—figure supplement 1 , profile 3)", make effect bigger in Figure 1—figure supplement 1 so it is clearer to see the differences between profile 3 and 7.

This has been revised.

Referring to "would be aggregated with nominal synergistic effects due to frequent cellular or assay saturation, which have low biological significance (see Figure 1—figure supplement 1 , profile 7)”. Please explain better the text before the comma. I think in profile 7 you are referring to the "floor" profile, where background in the assay or basal levels of gene expression makes it look like synergistic, and in your current explanation you are referring to the "ceiling" profile, where saturation is the problem. You can write something similar as it appeared later in the paper regarding these interactions that are likely not meaningful: "represent range limitations of the assay or biological responses"

This has been revised.

Multiple instances:Please replace TNF-α with simply TNF. The former term is obsolete (there is no TNF-β…) and has been retired.

This has been revised.

Figure 1: The panels for gene 2 and gene N could be permuted to maintain the hierarchy shown in the second column (synergistic > additive > antagonistic). The graphical rendition of drug interactions as slanted panel in the second column is hard to interpret: what do the authors want to highlight there?

We now clarify this in the figure legend. We accept the reviewer’s permutation suggestion and have revised this figure. The slanting in column 2 is to indicate that the slanted pattern is one of an entire deck of synergistic or antagonistic patterns classified by iSAIL.

Figure 2D: stimulat -> stimulate

This has been revised.

Figure 2E: Specify which IFN.

This has been revised.

Figure 5 legend: Include concentrations of IFN-β and TNF (these could be normalized to EC50 to make it easier to read and interpret -is the experiment done at saturation -ceiling-, at subdetectable levels -floor- or in linear regime; around EC50)

The concentrations have now been included.

Label the columns in panel c

This has been revised.

Reviewer #3:This study utilizes machine learning to examine gene expression datasets of immune stimulation and attempts to derive a universally applicable set of canonical responses. The basic approach dates back to the work of Janes et al. (Science 2005), which, incidentally, is not cited. My main concern centers on the need for a relatively large amount of data combined with synthetic data in order to obtain insights focused on well-studied interactions of TNF with other cytokines. Furthermore, I am unclear as to the true universal applicability of this approach, for the following reasons:

When analyzing combinatorial treatments datasets, there are two major challenges. One is classifying and interpreting the presence of interactions. The second is predicting the occurrence of interactions from single treatment data. From this reviewer’s citing of the Janes et al. study, we suspect that these two different goals (classifying and interpreting vs predicting interactions) have not been clearly distinguished in introducing the problem setting, and this may have caused a lack of clarity regarding the main contributions of our work. The Janes study was directed at predicting apoptotic responses from detailed signaling assays. In contrast, our study addressed a completely different problem: systematically detecting, classifying and interpreting interaction effects in a dataset and integrating insight across many studies. In this context, we clarify both the objective of the framework and derived biological insight below.

Objective of the iSAIL method: iSAIL is not a “prediction system”, i.e. our objective is not to predict, denovo, the occurrence of interactions. The goal of our method is to robustly identify, classify and interpret interactions. This work is a crucial advance, a statistically rigorous identification of interaction effects has been lacking, which has limited the insight gained from combination treatment experiments using heretofore available tools. Furthermore, the method is not predicated on the availability of many datasets (as possibly implied by the reviewer). To clarify, we generate a synthetic dataset to train a machine learning classifier – the only scenario where the ground truth classification of interactions is known. Once the model parameters are optimized, any individual dataset constructed according to the design of a standard combinatorial treatment experiment, including control, single and combination treatment conditions, can be analyzed using iSAIL.

Biological insight: The reviewer’s main concern is that the insights gained by iSAIL involve a well-characterized system, such as TNF in combination with other cytokines. Our validation system was not meant to confirm the generic prediction that TNF synergizes with another cytokine (e.g. IFN-β), as is well known in the literature. Rather, the validation aimed to show our ability to correctly infer the biological effects and mechanisms associated with such interactions. The validation experiments confirmed the validity of our inferences of biological mechanisms predicated on VCAM1 and CCL5 which are, to the best of our knowledge, original and previously unpublished results.

From this perspective, the ability of iSAIL to identify new biological effects and corresponding mechanisms of interactions in a relatively well characterized system can be seen -different from the reviewer's interpretation- as a proof-of-concept that our method holds an even greater potential for discovery in less characterized systems of signal combinations.

Finally, while we do not aim to directly predict interactions in individual datasets, we show that our framework does provide new insight on general rules that govern combinatorial interactions. As we state in the Discussion, these general rules can be used in future work aiming to predict interactions, but this possibility is tangential to the main research objective of our work. We acknowledge that our results on the probabilistic algebra may have generated confusion, as that section is thematically close to predictive modelling. Thus, in this revised, resource-oriented version, we have removed this section from the Results.

1) The authors mention that the datasets used involved multiple time points, but it is unclear how dynamics are being handled and modeled across these datasets. One would fully expect that in any extensive time course study of two stimuli, multiple outcomes (e.g. potentiation followed by tolerance or stimulation followed by suppression) would be observed. How can the algorithm then decompose such phenomena into a single characteristic response (i.e., does the algorithm treat each time point as a separate experiment, thereby losing true time-dependent interrelationships?)

We agree with the reviewer that the dynamics of signal integration is a very relevant, yet largely unexplored area. Extensive search of public data repositories (datasets deposited before 2019) showed that immunological combinatorial datasets with time course sampling are rare, precluding systematic research in this direction. Included in our compendium are time-course datasets with only two time points. Due to such poor temporal sampling, these studies were analyzed as separate experiments. While independent analysis of each time point would not be optimal for longer time series, developing a method to model temporal dynamics in combinatorial treatment experiments is impractical because of paucity of data.

Although the dynamics of combinatorial responses is not addressed in our work, our new analysis framework still provides a fundamental grammar and set of tools to map gene trajectories from longer time series. Time-dependent interrelationships could be included in our analysis, for example, by imposing additional constraints that guarantee smooth transitions between interaction profiles at successive time points. Our analysis framework, applied to a large compendium of time course experiments, should they become available, will enable us to identify preferential trajectories and dynamic rules governing time transitions between interaction profiles, similar to the probabilistic algebra we have identified in the static case.

Although the dynamics is a key aspect of combinatorial treatments, due to a complete lack of appropriately structured datasets and to the need for further extensive development, we consider it beyond the current scope of our work. To acknowledge the relevance of the dynamics, we have added a new paragraph to the Discussion.

2) The system chosen for validation, namely TNF + IFN-β, does not seem particularly novel. It has been appreciated for decades that TNF synergizes with other cytokines to produce an array of effects, including potentiation and tolerance as examples of preconditioning. A demonstration of a truly unexpected set of interactions between stimuli would have been more convincing.

As clarified in the response to the reviewer (under General assessment and major comments section), Our validation system was not meant to confirm the generic prediction that TNF synergizes with other cytokines (e.g. IFN-β), as is well known in the literature, including in our publications. Rather, the validation aimed to show our ability to correctly infer the biological mechanisms of such interactions. For more on this point, please refer to our answer above.

3) The additional validation studies focused on VCAM1 also seem to be focused on relatively well-studied phenomena. Multiple other gene transcripts that appear more novel are not followed up.

Our validation studies reveal important biological effects and mechanisms mediated by synergistic interactions between TNF and IFN β. To the best of our knowledge, these results are original and unpublished. The choice of VCAM1 and CCL5 as validation targets was driven by a combination of data-driven discovery, as provided by iSAIL, and prior knowledge. In our opinion, using prior knowledge, such as previously published literature and well-annotated functional pathway libraries at the validation step does not diminish the originality of our results. For example, while VCAM1 is generically known as a mediator of cell adhesion, no study has previously shown in a data-driven way that this gene is synergistically induced by TNF and IFNb, and validated the hypothesis that it controls the DC / T cell crosstalk.

When applied to a combinatorial treatment experiment of interest, iSAIL returns a set of transcripts displaying the most relevant interaction effects. These transcripts typically include both well- and poorly- characterized genes, both of which can be further explored in validation studies. The choice of gene candidates for further experiments depends on the context and underlying research objectives, and is ultimately the prerogative of the experimenter. In our immunological model, we focused on the discovery of new mechanisms regulating the crosstalk between DC and T cells, which is vastly recognized as one the most critical steps in the generation of an immune response.

**References:**

Mack, Chris A. "How to write a good scientific paper: acronyms." Journal of Micro/Nanolithography, MEMS, and MOEMS 11, no. 4 (2012): 040102.

Senthivel, Vivek Raj, et al. "Identifying ultrasensitive HGF dose-response functions in a 3D mammalian system for synthetic morphogenesis." Scientific reports 6.1 (2016): 1-15.